# Twenty-Four-Hour Movement Behaviors and Social Functions in Neurodiverse Children: A Scoping Review

**DOI:** 10.3390/bs15050592

**Published:** 2025-04-28

**Authors:** Chengwen Fan, Pan Liu, Zongyu Yang, Liqin Yin, Shuge Zhang

**Affiliations:** 1College of Physical Education, Hunan University of Technology, Zhuzhou 412007, China; 2Centre for Sport and Psycho-Social-Behavioral Research, Zhuzhou 412007, China; 3College of Physical Education, Hunan Normal University, Changsha 410012, China

**Keywords:** neurodiversity, children, 24 h movement behavior, physical activity, sedentary behavior, sleep, social responsiveness, social function

## Abstract

Research on how an active lifestyle impacts the social functions of neurodiverse children, particularly within the context of twenty-four-hour (24 h) movement behaviors (i.e., physical activity, sedentary behaviors, sleep), has been emerging but has yet to receive a systematic synthesis. In this scoping review, we aimed (1) to synthesize current knowledge in the field of 24 h movement behaviors and social functions in neurodiverse children; and (2) to offer insights into implications for future research and practices. Specifically, we conducted a systematic search via four databases, namely the Web of Science, PubMed, Scopus, and EBSCOhost, through 31st December 2024, and followed scoping review guidelines for results synthesis. The initial search returned 2342 articles, of which 50 studies fulfilled the inclusion criteria following a robust selection and screening process. These retained studies were published between 2004 and 2024, primarily focused on children with neurodiversity of autism spectrum disorder, or ASD (70%), and attention deficit/hyperactivity disorder, or ADHD (23%), and using quantitative methods (84%). Only 6% of studies considered the combined effects of different 24 h movement behaviors, with most examining physical activity alone. Longer group exercise interventions were found to be more effective in improving social functions in neurodiverse children. Overall, the findings support the position that 24 h behaviors have a positive influence on the social functioning of neurodiverse children. However, current research tends to focus primarily on ASD, uses quantitative methods, and often overlooks the combined effects of physical activity, sleep, and sedentary behavior. Future studies should address these limitations and examine 24 h movement behaviors in children with a broader range of neurodiverse characteristics. Research and practices should also consider qualitative approaches as a complement to quantitative measures for monitoring and evaluation.

## 1. Introduction

Mental health among children and adolescents is a global public health issue, with one-quarter of this population affected by various mental disorders ([24]). However, the risk of experiencing mental health issues is even greater among children and adolescents with neurodevelopmental disorders (NDDs), such as autism spectrum disorder (ASD), attention deficit/hyperactivity disorder (ADHD), and other atypical cognitive or neuropsychological characteristics or disorders rooted in one’s early development ([40]). Indeed, children with NDDs tend to delay the development of cognitive-related social functions, demonstrating interpersonal difficulties, behavioral problems, and deficits in executive function ([13]; [16]). Such challenges impede their ability to engage in typical social interactions, such as communication with parents and peers ([27]), which in turn contribute to maladaptive emotional regulation and impaired mental health ([19]). Population-based studies have revealed that approximately 52% of children and adolescents with ADHD ([16]) and 27–95% of those with ASD ([43]) experience varying degrees of mental health problems. Evidence supports the benefits of an active lifestyle for enhancing general mental health and reducing difficulties related to social function and behavioral challenges in neurodiverse children ([20]). However, research has yet to provide conclusive evidence on the extent to which an active lifestyle helps improve social functioning in neurodiverse children (e.g., social interactions and behavioral problems). Such a perspective into social functioning in neurodiverse children is critical because it shadows emotional regulation and many related aspects that underpin mental wellbeing in neurodiverse children. Therefore, we conducted a scoping review that aimed to provide a wide evidence synthesis on the effect of an active lifestyle on neurodiverse children’s social function.

To expand, neurodiversity refers to the cognitive and emotional traits associated with developmental disorders, such as ASD, in which subjects may appear “impaired”, but they are not necessarily deficits. In other words, rather than seeing developmental issues (e.g., ADHD, ASD) as a disorder or illness, the conceptualization of neurodiversity posits that the so-called developmental issues are normal human cognitive–behavioral variations underpinned by varied personal-level characteristics and dispositions born to life ([2]). Such a conceptual approach rejects the “traditional” practice of viewing these children as patients or disabled ([22]). In other words, neurodiversity is not necessarily a clinical concept; the manifestations of neurodiversity traits or characteristics may well exist in subclinical or nonclinical populations.

In addition to the commonly known mental health challenges in neurodiverse children ([19]), researchers have generally agreed on the benefits of physical activity in such contexts. Specifically, a recent meta-analysis ([19]) synthesized evidence from 52 randomized controlled trials and 24 non-randomized studies, involving a total of 3007 participants aged 5–17 years with ADHD (n = 31 studies), ASD (n = 19 studies), and other NDDs (n = 26 studies). The findings provided support for the effectiveness of physical activity interventions in improving the mental health of children with NDDs. Regardless of the positivity in its findings, the work only looked into general aspects of mental health in children with NDDs but overlooked their social–behavioral underpinnings (e.g., social and interpersonal functions or abilities). Such a gap could undermine the design and efficacy of physical activity-related mental health interventions for children with NDDs, as knowledge is scarce about the social–behavioral mechanisms of physical activity in children with NDDs; thus, it is unlikely that effective monitoring and evaluation will be implemented to understand why an intervention is (in)effective. Examining the role of physical activity in the social and interpersonal functions of children with NDDs is, therefore, central to informing the design and delivery of physical activity-based interventions for children with NDDs.

With this critical research agenda in mind, emerging research has started to apply interventional ([11]; [14]; [28]) or cross-sectional ([7]; [21]; [38]; [39]; [47]) methods to examine the effectiveness of physical activity in enhancing social or interpersonal functions of neurodiverse children. Despite these efforts, the lack of a wide, systematic synthesis of evidence in this domain is noteworthy. We argue that the collation of such evidence should not be restricted to quantitative formats, because qualitative and mixed-methods research of physical activity in the contexts of social functions and neurodiverse children also provides valuable implications ([38]). As such, in the present study, we aimed to conduct a scoping review that collates diverse evidence (e.g., interventional, observational, qualitative, or mixed methods) regarding active lifestyle and social functioning in neurodiverse children.

Further, as a novel and significant contribution, the present scoping review extended physical activity as a key health and movement-related behavior by embracing and incorporating the concept of 24 h movement behavior. In its conceptualization, 24 h movement behavior consists of physical activity (PA), sedentary behavior (SB), and sleep (SP) throughout an individual’s typical daily life ([41]). Such a concept is desirable and increasingly adopted in health behavior research because the three different movement behaviors constitute one’s 24 h daily cycle, so that an increase (or decrease) in one behavior inevitably leads to a decrease (or increase) in the other two movement behaviors, thus amplifying or alleviating health impacts ([46]). Despite being a relatively new concept, 24 h movement behavior has been adopted in research on social functions in children with NDDs. For example, research has found that sleep problems in children with ADHD, regardless of physical activity, are associated with the deterioration of social functions ([7]). Cross-sectional studies have also demonstrated evidence that adhering to 24 h movement behavior guidelines is generally associated with better social cognition, social functions, and emotional health in children and adolescents with ASD, ASD + ADHD, or learning disabilities (LD) ([21]; [39]; [47]). Given this research trend (i.e., adopting 24 h movement behaviors) and due to the lack of research findings synthesis, we extended our scoping review by going beyond physical activity and included the other two movement behaviors (i.e., sedentary behavior, sleep) for a fuller picture of how an active lifestyle contributes to social functions in children with neurodiversity. Overall, this review aimed to generate a wide synthesis of evidence that provides insights into how 24 h movement behaviors affect social functioning in neurodiverse children.

## 2. Materials and Methods

We conducted the review following Arksey and O’Malley’s guidance for scoping reviews ([1]). To align with the recent development of scoping review methods, we also consulted updated scoping review methodology proposed by [8] ([8]), [18] ([18]), [30] ([30]), and [31] ([31]). More specifically, we adhered to Arksey and O’Malley’s five key stages of the scoping review process in principle, including *(1) defining the research question*, *(2) identifying relevant studies*, *(3) selecting studies for inclusion*, *(4) extracting and charting data, and (5) synthesizing and summarizing findings* ([8]; [18]; [30]; [31]). The review protocol was pre-registered on the Open Science Framework (see: Appendix A). Below, we report the scoping review methods and results following Expansion of Scoping Reviews for Systematic Evaluation and Meta-Analysis (PRISMA-ScR) guidelines ([42]).

### 2.1. Information Sources and Search Strategy

The databases searched included the Web of Science, PubMed, Scopus, and EBSCOhost. covering studies from their inception to December 2024, with a final search conducted in January 2025; thus, data completeness was ensured for the targeted timeline of potential studies to be included for this review. Keywords and text words contained in titles and abstracts of relevant articles were used to develop full search strategies for databases. The strategy incorporated four sets of keywords: “24-h movement behavior”, “neurodiversity”, “social function”, and “children”. Search terms consisted of subject terms and free-text words related to the four key concepts above. To enhance the search process, we use a number of Boolean operators (AND, OR) with field identifiers or qualifiers (TI, TS, *). For example, the full search formula in the Web of Science is as follows: TS = (24-h movement behavior OR 24-h movement OR 24-h activity OR physical activity OR exercise OR sedentary behavior OR physical inactivity OR screen time OR sleep OR insomnia OR sleepless) AND TS = (neurodiverse OR neurodiversity OR developmental disorders OR mental disorder OR ADHD OR attention deficit hyperactivity disorder OR ASD OR autism OR autism spectrum disorder OR developmental language disorders OR dyslexia OR dyscalculia OR aphasia OR learning difficulties) AND TS = (Social functioning OR community functioning OR social competence) OR TS = ((social OR interpersonal) AND (skill OR interaction OR dynamics)) AND TS = (Child* OR kid* OR boy* OR girl* OR schoolchild* OR early-years OR preschool OR school). Additionally, we screened the references of the included studies for any eligible records. More detailed search strategies and our step-by-step use of each database are available in Appendix A of the study pre-registration (see: Appendix A).

### 2.2. Eligibility Criteria

We included studies of neurodiverse children and adolescents (under 18 years) that investigated the effect of 24 h movement behavior on social functioning. PA, SB, and sleep during the day constituted the 24 h movement behavior ([35]). It is important to note that studies drawing from the results of our first search containing all three or at least two 24 h behaviors was rare. Therefore, we decided to include studies that included one or more activity behaviors. At the same time, we included children with at least one neurodiverse characteristic (such as children with autism spectrum disorder, attention deficit/hyperactivity disorder, learning disabilities, etc.) as the main population of the study, regardless of whether they were diagnosed or not. In addition, the social functioning of neurodiverse children was used as an outcome indicator for the study. It is noteworthy that the studies we included were empirical studies (e.g., cohort studies, cross-sectional studies, intervention trials, and longitudinal studies). As such, non-empirical studies (e.g., research reviews, conferences, reports, etc.) were excluded. In addition, studies in which the primary population was one that did not exhibit neurodiversity characteristics or was 18 years of age or older were excluded. Similarly, studies that did not include any of the 24 h movement behaviors were excluded. We also excluded studies that were not published in English-language refereed journals. More details of the eligibility criteria are available in Appendix A the study pre-registration (see: Appendix A).

### 2.3. Screening

All retained studies were included in a tool called the Systematic Review Accelerator (SRA) online to screen the literature for duplicate content ([3]), available at https://sr-accelerator.com/#/ (accessed on 3 January 2025). We also used this tool to exclude non-English and unrelated records. After that, all records were imported into EndNote X9 software (Thomson Research Soft, Stanford, GA, USA) for review and screening. Three junior researchers independently screened the titles and abstracts of the identified records, followed by full-text screening of potential studies to determine eligibility for inclusion in this scoping review. Disagreements were resolved through discussion with two senior researchers until a consensus was reached.

### 2.4. Data Extraction and Synthesis of Results

Data for final inclusion in the study were extracted using an Excel spreadsheet. The extraction of data was done by one researcher, and another researcher checked the accuracy of the information extraction; if there was any disagreement, the senior researchers were involved in a discussion. For each selected study for the review, we extracted content related to the *study population* (neurodiversity, age, and sample size), *study characteristics* (study design and location), *intervention* (if applicable; content of intervention, duration/frequency, and sample size of intervention and control groups), the *main research question*, as well as *key findings*. To present our results succinctly and align with the central objectives of this review, we chose to report only the primary social functioning outcomes, as the aim of this scoping review is to focus on the impact of 24 h movement behaviors on social functioning specifically, rather than exploring a broader range of outcome indicators beyond social function-related variables. Beyond this, we report only published studies and do not consider ongoing studies. Given the nature of scoping reviews ([1]) and the PRISMA-ScR guidance ([42]), we did not perform quantitative analysis on the extracted data, nor did we assess the quality of the included studies, but focused on synthesizing findings and creating a knowledge structure of existing research regarding 24 h movement behaviors and social functions in children with neurodiversity.

## 3. Results

### 3.1. Study Selection

Following an updated search in December 2024, the details of the study screening and inclusion are outlined in Figure 1. Specifically, 2342 records were initially identified, of which 1080 were removed due to duplication (i.e., representing 288 unique articles). Of the remaining 1262 studies, 1088 were excluded during the title and abstract screening, leaving 174 studies for full-text review. These studies were then imported into EndNote X9 for full-text review. After reviewing the full texts, 130 studies were excluded, and 6 additional studies were included by tracking the references of the initially included 44 studies. Thus, a total of 50 studies were included in this review (see Figure 1).

### 3.2. Study Characteristics

The publication years of the included studies span from 2004 to 2024, with a noticeable increase in the number of publications over time (see Figure 2). Table 1 provides comprehensive details regarding the characteristics of the studies included in this review. The majority of the studies are quantitative in nature, consisting of observational research (24%), intervention or experimental research (60%), and mixed-methods research (16%). Among these, the distribution includes cross-sectional studies (18%), quasi-experimental studies (30%), mixed-methods studies (16%), and randomized controlled trials (26%), while a smaller proportion includes case-controlled, prospective case-control, NCT, and mixed experimental designs. Notably, the studies selected for inclusion did not employ longitudinal methodologies or exclusively qualitative approaches. The majority of studies focused on the relationship between physical activity (PA) and the social functioning of neurodiverse children (76%), followed by studies examining the association between social participation (SP) and social functioning (16%). A small subset of studies (6%) simultaneously explored PA, sedentary behavior (SB), and SP in relation to the social functioning of neurodiverse children.

The included studies primarily originated from the United States (22%) and China (20%), with additional contributions from Iran (14%), Turkey (8%), and Australia (6%) (see Table 2). These studies are geographically diverse, representing a broad global distribution. It is important to note that the classification of a study’s country of origin was based on the location of the sample population rather than the country of the primary researchers, unless the origin of the sample population was not reported. This approach was taken to assess potential differences in the level of attention devoted to neurodiverse children across different regions.

### 3.3. Characteristics of Participants

Table 2 presents full details of the characteristics of the included study samples. The majority of study participants were diagnosed with ASD (70%), followed by those diagnosed with ADHD (16%). In addition, other types of neurodiversity accounted for a smaller proportion (e.g., developmental disabilities, ASD + ADHD). The mean age of children in the included studies was primarily 7–11 years (52%), followed by 2–7 years (30%), while 18% of the studies reported an average age of 11–18 years. Among these participants, 70% of the studies included both boys and girls, 16% recruited only boys, and 14% did not report gender characteristics. The neurodiverse children included in the study were predominantly from the United States (22%) and China (20%).

### 3.4. Study Measures

This study primarily investigates the measurement of 24 h movement behaviors (i.e., PA, SB, SP), neurodiversity (e.g., ASD and ADHD), and social functioning. Among the studies reviewed, only 2% assessed children’s PA using the *Child Behavior Checklist* (CBCL), and another 2% employed the GT3X Actigraph monitor to measure PA. A substantial proportion of studies (74%), however, did not report the methods used to assess PA. Regarding the measurement of SP, the two most frequently used tools were the *Children’s Sleep Habits Questionnaire* (CSHQ), employed in 14% of studies, and the *Pediatric Sleep Questionnaire* (PSQ), used in 4%. SB was primarily assessed through parental reports in 2% of studies. Neurodiversity traits were assessed using the *Diagnostic and Statistical Manual of Mental Disorders* (DSM) in approximately 48% of studies, followed by the *Gilliam Autism Rating Scale* (GARS), used in 12% of studies. In terms of evaluating social functioning, the two most commonly employed instruments were the *Social Responsiveness Scale* (SRS), used in 16% of studies, and the CBCL, used in 8% of the included studies.

Next, we present the results based on different neurodiversity profiles (categorized into ASD, ADHD, and other neurodiverse conditions), as shown in Table 3. Among the 35 studies focusing on children with ASD, the majority (89%) did not specify the method used to assess PA. Only two studies used the CBCL and the GT3X Actigraph monitor, respectively, to measure PA. Furthermore, one study assessed SP using the CSHQ, while another relied on parental subjective reports to measure SB. The most frequently employed tools for measuring neurodiversity traits in children with ASD were the DSM (49%) and the GARS (17%). Twenty percent of the studies used the Social Responsiveness Scale (SRS) to assess social functioning, while 11% utilized the GARS. Among the eight studies focused on children with ADHD, three did not report the method used to assess PA. Notably, five studies (64%) concentrated on SP, with half of these (50%) using the CSHQ. No studies measured SB in children with ADHD. The DSM was the primary tool for assessing neurodiversity traits in children with ADHD, appearing in 63% of studies. Additionally, 60% of studies utilized the Strengths and Difficulties Questionnaire (SDQ) to assess social functioning. Finally, seven studies focused on other forms of neurodiversity, with 71% failing to report the method used to assess PA. Among the remaining studies, two used the CSHQ and the PSQ, respectively, to assess SP. The DSM was the main tool for measuring neurodiversity traits in 29% of these studies, while the methods for assessing social functioning varied.

### 3.5. Design of Interventions

Among the 50 studies ultimately included, 86% involved interventions, which varied in their approaches. Of the 35 studies focusing on children with ASD, 89% employed experimental designs (or mixed methods). Of these, 68% used group-based activities (e.g., basketball, soccer) for the intervention, while 22% used structured physical activities (e.g., yoga, martial arts). In 60% of the studies, the total intervention duration was no less than 12 weeks, and 58% had an intervention frequency of at least three times per week. Additionally, 90% of the studies reported an intervention duration of at least 40 min per session. These interventions showed a significant improvement in children’s social functioning. Notably, only one study evaluated the long-term effects of the intervention, assessing outcomes beyond six months ([45]). Of the eight studies focusing on children with ADHD, four employed experimental designs. Sleep interventions (50%) and individual physical activities (50%) were the primary intervention methods. All studies had a total intervention duration of no less than 12 weeks, with 50% having an intervention frequency of at least once per week. Additionally, 38% of the studies reported a minimum duration of 30 min per intervention session. These interventions significantly improved children’s social functioning, particularly in terms of emotional and behavioral issues. Among the seven studies on other forms of neurodiversity, three used experimental designs (or mixed methods), with group-based activities and family involvement as the primary interventions (100%). In 67% of the studies, the total intervention duration was no less than 8 weeks, and 67% had an intervention frequency of at least once per week. All studies reported that the intervention sessions lasted at least 60 min. These interventions also showed a significant improvement in children’s social functioning.

### 3.6. General Overview of Scoping Review Findings

We present a summary of key information from each included study in Table 3. Overall, more than 80% of the studies found that regular activity behaviors (adequate physical activity and sleep but limited sedentary behavior) significantly improved the social functioning of neurodiverse children, especially by enhancing social interactions (56%), reducing feelings of loneliness (2%), and strengthening emotional regulation abilities (16%) ([12]; [21]; [23]; [33]; [39]). Another important aspect is that, among the nine mixed-methods studies, five studies (10% of the total included studies) did not find positive effects on children’s social functioning in the quantitative results, while the qualitative results confirmed this finding. To present our results more concisely, we will present a detailed report based on children’s neurodiversity characteristics (ASD, ADHD, and other forms of neurodiversity).

**Social functioning of children with ASD**. Among all the studies focusing on children with ASD, 80% were quantitative studies (RCT = 37%, quasi-experimental = 34%, cross-sectional = 9%), and 20% employed mixed methods (mixed methods = 14%, mixed experimental design = 6%). In the quantitative studies, 40% of the studies reported that physical activity interventions improved social skills, 6% of the studies found that sleep interventions or improved sleep positively impacted children’s social skills, and one study revealed that reducing screen time (i.e., sedentary behavior) had a positive effect on the social skills of children with ASD. Additionally, 20% of the studies reported that physical activity interventions reduced behavioral problems in children with ASD, one study found that improved sleep quality alleviated behavioral issues, and one study showed that reducing screen time decreased the incidence of behavioral problems. Among the mixed-methods studies, 11% of studies indicated that physical activity interventions enhanced communication abilities in children with ASD. It is important to note that, although 11% of the studies did not report positive effects in their quantitative results, their qualitative findings consistently indicated beneficial impacts on improving social functioning ([4]; [10]; [34]; [36]). In addition, none of the studies adopted a longitudinal follow-up design or were purely qualitative. In the research on children with ASD, 34% of the studies included participants with an average age of 2–7 years, 51% of the studies focused on participants with an average age of 7–11 years, and 6% of the studies involved participants with an average age of 11–18 years.

**Social function of children with ADHD**. Among the studies included in the final analysis, 23% focused specifically on children with ADHD. Of these, 50% employed experimental designs (RCT = 38%; quasi-experimental = 13%), while 50% were cross-sectional studies. The intervention studies generally found that both sleep and physical activity interventions improved social skills, behavioral problems, and executive functioning in children with ADHD. Specifically, sleep interventions were more effective in improving emotional regulation, behavioral issues, and executive functioning, whereas physical activity interventions primarily enhanced social and behavioral outcomes. In addition, two cross-sectional studies reported significant associations between sleep problems and social interaction or emotional difficulties in children with ADHD. One cross-sectional study found that insufficient daytime sleep was associated with social difficulties. Unlike for children with ASD, sleep emerged as a key intervention or associated factor for improving social functioning in children with ADHD (n = 63%). Notably, none of the studies focusing on children with ADHD employed a longitudinal follow-up design, mixed methods, or purely qualitative approaches. Regarding the age of participants, 75% of the studies included children with an average age of 7 to 12 years, and no studies included preschool children (under 6 years old).

**Social function of children with other neurodiversity**. Of the studies included in the final analysis, 14% focused on other types of neurodiversity, with 6% examining developmental disorders, 4% addressing both ASD and ADHD, 2% exploring neurodevelopmental disorders, and 2% investigating learning disabilities. Among these studies, 29% were cross-sectional, 29% utilized mixed methods, and 14% comprised randomized controlled trials, prospective case-control studies, and retrospective clinical trials, respectively. Overall, the studies consistently found that sleep problems were associated with social skills and behavioral issues, while physical activity interventions were beneficial in improving social functioning and emotional regulation in neurodiverse children. Additionally, some studies reported that adherence to 24 h exercise behavior guidelines was positively associated with improved social functioning and emotional wellbeing in neurodiverse children. Notably, quantitative results from one mixed-methods study showed no improvement in social behavior following exercise interventions, although qualitative findings indicated an enhancement in psychological resilience. Furthermore, these studies did not address the importance of longitudinal follow-up or purely qualitative methods. Finally, 71% of these studies focused on children aged 6 to 12 years.

## 4. Discussion

Neurodiverse children are more prone to mental health problems ([40]). One significant factor that contributes to mental illness in neurodiverse children is their atypical cognitive and neuropsychological development in social functions ([40]). Building on existing knowledge of physical activity and social functions, this scoping review provided an up-to-date knowledge synthesis regarding 24 h movement behavior and social functions in children with neurodiversity characteristics (e.g., ASD, ADHD). The emphasis of 24 h movement behavior ([41]) extended the classic physical activity concept to a recent development in the field by encompassing not only physical activity but also sedentary behavior and sleep, thus offering a fuller picture of one’s active lifestyle. Following a rigorous procedure for conducting this scoping review, we found that over 80% of the 50 included studies showed that balanced movement behaviors, such as sufficient physical activity, adequate sleep, and reduced sedentary time, significantly improved social skills and emotional regulation and reduced loneliness in neurodiverse children (i.e., mainly ASD or ADHD). However, most existing studies were observational, with few utilizing randomized controlled or longitudinal designs. Whilst most included studies adopted quantitative methods, five took advantage of the mixed-methods designs, providing qualitative insights beyond quantitative ones. We highlight several noteworthy points and key takeaways and implications for practice from the scoping review findings below.

### 4.1. Strength of Evidence and Moving Beyond Physical Activity

We found that 94% of the studies considered only one type of 24 h movement behavior (PA, SP, or SB). The vast majority of these studies focused primarily on physical activity (81%), of which most employed experimental designs (92%). Conclusions drawn from these studies were generally consistent between different study designs (i.e., experimental vs non-experimental) and between different movement behaviors. Specifically, strong evidence existed supporting the role of physical activity in improving neurodiverse children’s social function, particularly in enhancing social skills and emotional and behavioral regulation. It is worth noting, however, that only a small number of studies have examined the impact of sleep (16%) and sedentary behavior (2%) on the social functions of neurodiverse children. These less-representative studies also support the position that a healthy 24 h movement behavior (e.g., increased sleep and reduced sedentary behavior) is beneficial to children with neurodiversity. Future research should consider examining these two important 24 h movement behaviors in the contexts of neurodiverse children and social functioning for further replication and extension.

Furthermore, it is also noteworthy that the concept of 24 h movement behavior highlights the critical interplay between physical activity, sedentary behavior, and sleep throughout a typical 24 h daily cycle ([41]). The combined effects of these behaviors, therefore, contribute to one’s health and wellbeing more prominently than one single movement behavior ([5]). However, only 6% of studies in this review examined the combined effects of different 24 h movement behaviors. With this in mind, we call for future reviews focused on neurodiverse children to adopt a more comprehensive approach by collecting data on multi-dimensional 24 h movement behaviors (i.e., considering at least two different movement behaviors). Such an approach could offer insights into the efficacy of more integrated movement behavior interventions compared to more traditional interventions that only address physical activity, sedentary behaviors, or sleep. Practitioners (e.g., special education teachers, therapists, coaches) should consider not only increasing physical activity levels but also promoting enhanced sleep and a reduction in sedentary behavior to enhance social functioning in neurodiverse children.

### 4.2. Optimizing Intervention Designs for Neurodiverse Children

This review also unveiled that group-based exercise (71%) and individual exercise (21%) were the main intervention forms for enhancing social functions in neurodiverse children. To unfold this result, the findings of studies utilizing group-based exercise were generally consistent and positive (i.e., exercise improved the social functions of neurodiverse children). These studies appeared to have a more pronounced effect on reducing social withdrawal and feelings of loneliness. For comparison, the findings of studies utilizing individual-based exercise were also consistent but appeared to enhance self-regulation skills and social competencies more prominently. These findings are in line with a recent meta-analysis of group-based exercise in children with ASD ([15]) and extend the literature by providing insights into the comparison of group vs individual exercise settings and how different settings affect social functions in children with neurodiversity (i.e., beyond ASD). Indeed, group exercise settings provide greater opportunities for social interaction and communication, but individual exercise settings may be better for repetitive motor skill training that activates exercise-related brain regions and facilitates cognitive development ([32]). As such, practitioners would do well to consider different exercise settings (i.e., group vs individual) for different aspects of social functions and the related mental health outcomes when designing and delivering interventions for neurodiverse children.

Another critical factor that affects intervention effectiveness in the social function of neurodiverse children is the actual time involved in the intervention. Our review revealed that the average intervention length was 14.5 weeks, with most studies lasting over 12 weeks (79%), whilst the shortest was 5 days (i.e., [37]). Interestingly, the reported 5-day intervention also significantly improved intervention outcomes (i.e., social skills, self-esteem). Nevertheless, session length appeared to play a more critical role; that is, a longer session length (especially for those tackling physical activity) was associated with more desirable outcomes. For context, intervention studies included in the current review were operated for at least 40 min in guided exercise tasks. Since exercise load, volume, and frequency influence the outcome related to exercise ([9]), a longer session length is more feasible for implementing the desired exercise and thus provides greater benefits ([19]). However, it is unlikely that longer sessions or interventions would always lead to more desired outcomes. For instance, longer sessions or extended interventions may lead to physical and mental fatigue in children, reducing their enthusiasm. Therefore, we call for practitioners to implement at least 40 min exercise tasks per session when designing and delivering an exercise intervention for children with neurodiversity. Future researchers would do well to examine whether a dose–response relationship exists for 24 h movement behavior and social functions in neurodiverse children.

### 4.3. Taking the Advantages of Qualitative Measures

Moreover, we found that quantitative research (84%) constituted the majority of the studies in this review, with only a few utilizing a mixed-methods design (16%). No study exclusively employed qualitative methods. Since the improvement in the social functioning of neurodiverse children may result from multiple factors ([6]), quantitative measures may not always capture changes in social function outcomes. Our review uncovered that, in the five mixed-methods studies ([4]; [10]; [25]; [34]; [36]), quantitative measures did not improve at post-intervention, but qualitative measures (e.g., parent report or reflection) did. For example ([34]), a mixed-methods study conducted in the United States found that after an 8-week judo intervention, quantitative results indicated no significant improvement in the negative behaviors of children with ASD (i.e., irritability, lethargy, inappropriate behaviors, and hyperactivity), as there were no significant changes in the social function measures. However, qualitative data from parent interviews revealed that 78% of parents believed their children showed significant improvements in social skills, such as peer interactions, communication with family members, and overall social engagement. This highlights the value of qualitative methods in supplementing and expanding on improvements that quantitative data may overlook. Given the benefits of mixed-methods designs that combine the strengths of both quantitative and qualitative approaches ([17]; [26]), we call for more future intervention research of children with neurodiversity to consider mixed-methods designs, taking advantage of both quantitative and qualitative methods. Practitioners would also do well to consider qualitative measures for monitoring and evaluation, at least in the contexts of social functions and neurodiverse children.

### 4.4. Breaking the Stigma of “Disorder" or “Disability"

Last but not least, neurodiversity (e.g., ASD, ADHD, etc.) was considered a disability or disorder in the clinical paradigms ([14]). However, with the emergence of new perspectives, neurodiversity has been increasingly seen as a reflection of the broad range of diversity of personal characteristics within human neurobiology ([17]), rather than a disorder, disease, or disability. In the meantime, neurodiversity has been conceptualized in comparison to neurotypical traits or attributes, linking it to a natural variation in one’s cognitive and behavioral tendency and being rooted in the inherent individual characteristics or personality ([22]). Importantly, emerging research has supported the unique strengths of individuals with neurodiversity, such as creativity in children with autism ([29]) and cognitive flexibility in children with ADHD ([44]). However, all studies included in this review adopted a clinical population that was diagnosed with certain neurodiversity. Our concern regarding a sole focus on a diagnosed or clinical population in existing studies around 24 h movement behavior and social functions in neurodiverse children is two-fold. First, levels of neurodiversity can vary among individuals, and those who are not diagnosed with a certain neurodiversity may be at risk regarding their social functions or other related mental health issues due to higher-level neurodiversity characteristics despite not being diagnosed. Second, delivering interventions among diagnosed individuals implies that such interventions are used for treatment purposes, which overlooks the probably more important perspective that enhancing 24 h movement behavior could be used as a preventive strategy in individuals high in neurodiverse traits but yet to be diagnosed. Future research should consider extending current findings to the subclinical or normal population and consider assessing neurodiversity as a relative (i.e., akin to personality measures that can be quantified on a continuum), not absolute (i.e., diagnosed vs not diagnosed) construct. Practitioners, especially coaches or behavioral therapists, as well as other stakeholders (e.g., school managers, physical activity program providers, parents) would do well to acknowledge the existence of neurodiversity traits or characteristics in normal children and design movement-based programs (e.g., increasing physical activity, enhancing sleep, reducing sedentary behaviors) to attenuate the negative effects of neurodiversity. Thus, such programs could also serve a preventive function.

### 4.5. Strengths, Limitations, and Future Directions

This scoping review has several strengths. First, we expanded the scope to include children with a variety of neurodiverse traits, not limited to ASD or ADHD, and considered full-day movement behaviors (PA, SB, and SP). Second, we included both quantitative and qualitative studies, providing new perspectives to the existing literature. Lastly, we conducted a rigorous review process, including clear definitions of terms and a comprehensive, systematic search strategy.

However, several limitations should be noted. First, our primary objective was to examine the impact of 24 h movement behaviors on the social functioning of neurodiverse children. Although most studies assessed one of the three behaviors, only a few focused on the combined effects of all three. The concept of 24 h movement behavior takes a more comprehensive approach by considering the overall health benefits of the interaction between physical activity, sleep, and sedentary behavior, rather than just the benefits of physical activity alone, as sleep and sedentary behaviors are equally important ([5]). From this perspective, we recommend that future research should more comprehensively evaluate the effects of 24 h movement behaviors on the health of neurodiverse children, and not limit this approach to review studies. Secondly, the majority of the studies included in our review focused on children with ASD, followed by children with ADHD. However, our goal was to consider a broader range of neurodiverse traits in children, highlighting a gap in current research regarding other types of neurodiverse children. This may be due to the lack of theoretical frameworks addressing the impact of 24 h movement behaviors on the social functioning of neurodiverse children. Therefore, we recommend that future research should consider a wider variety of neurodiverse traits and examine whether enhancing 24 h movement behavior benefits social functions. Lastly, considering the nature of our scoping review (which focuses on providing a broad synthesis of evidence to help researchers understand the progress in this area, rather than quantifying the magnitude of a specific effect), we did not assess the quality of the included studies. However, we recommend that future studies consider quality assessments of the included research, as this would better address potential heterogeneity in the studies. Additionally, future reviews would do well to acknowledge potential publication bias and assess it accordingly, especially when studies reporting null or negative findings are underrepresented.

## 5. Conclusions

This scoping review synthesized existing literature evidence regarding the role of 24 h movement behaviors in the social functions of neurodiverse children. Whilst research to date has supported the benefits of healthy 24 h movement behavior (i.e., physical activity, sedentary behavior, sleep) on the social functions of children with neurodiversity, most of the research had a sole focus on physical activity rather than accounting for the combination of two or more 24 h movement behaviors. Stakeholders, such as school managers, exercise program providers, and parents, would do well to consider movement-based interventions or practices beyond increasing physical activity levels (e.g., enhancing sleep, reducing sedentary behaviors). Practitioners, such as teachers, therapists, and coaches, when working with neurodiverse children or children high in neurodiverse traits, should consider utilizing group- and individual-based activities, considering the benefits of different aspects of social functions in neurodiverse children. When delivering physical activity-based interventions, they should promote longer session lengths (i.e., >40 min), thus optimizing the effectiveness of the interventions. Future research and practice should consider involving the subclinical population with varied levels of neurodiversity traits and incorporating a mixed-methods design or qualitative measures assessing social functioning and other mental health-related outcomes in neurodiverse children.

## Figures and Tables

**Figure 1 behavsci-15-00592-f001:**
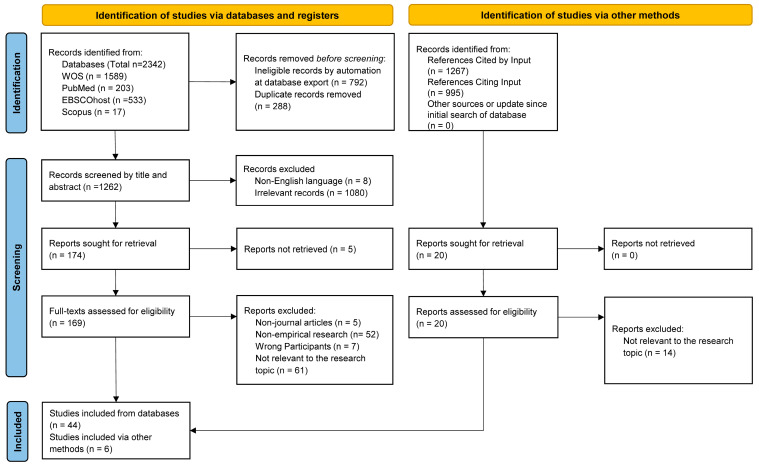
PRISMA flow diagram outlining study identification and screening process.

**Figure 2 behavsci-15-00592-f002:**
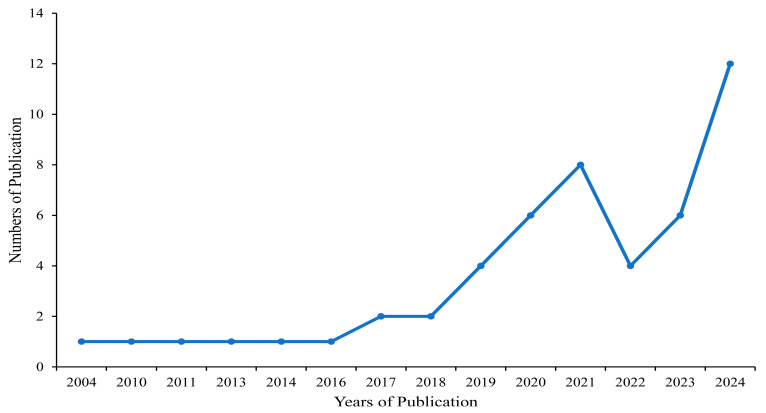
Number of publications per year of the included studies.

**Table 1 behavsci-15-00592-t001:** Summary of study characteristics included in scoping review.

Study Characteristics	N	% of Studies Sampled
Types of research		
Observational research	12	24.0
Intervention/experimental research	30	60.0
Mixed-methods research	8	16.0
Design		
Cross-sectional	9	18.0
Quasi-experimental	15	30.0
Mixed methods	8	16.0
RCT	13	26.0
NCT	1	2.0
Case-controlled	1	2.0
Prospective case-control	1	2.0
Retrospective clinical	1	2.0
Mixed experimental design	1	2.0
Measure		
Physical activity only	38	76.0
Sleep only	8	16.0
Sedentary behavior only	1	2.0
PA + SP + SB	3	6.0

Note. RCT = randomized controlled trial; NCT = non-randomized controlled trial; PA = physical activity; SB = sedentary behavior; SP = sleep.

**Table 2 behavsci-15-00592-t002:** Summary of study sample demographics included in scoping review.

Demographic Information	N	% of Studies Sampled	Demographic Information and Measures		N	% of Studies Sampled
Nationality of study sample			Mean age of study sample *			
Australia	3	6.0	2–7 years		15	30.0
Brazil	1	2.0	7–11 years		26	52.0
Canada	1	2.0	11–18 years		9	18.0
China	10	20.0	Gender of participants			
Egypt	1	2.0	Boys and girls		35	70.0
India	2	4.0	Boys only		8	16.0
Ireland	1	2.0	No report		7	14.0
Iran	7	14.0	Key measures for neurodiversity **			
Italy	2	4.0	DSM		24	48.0
Korea	2	4.0	GARS		6	12.0
Spain	1	2.0	Key measures for social function **			
Switzerland	1	2.0	SRS		8	16.0
Turkey	4	8.0	CBCL		4	8.0
Tunisia	1	2.0	Key measures for movement behavior			
USA	11	22.0	PA	CBCL	1	2.0
U.K.	2	4.0		GT3X monitor Actigraph	1	2.0
Neurodiversity				WOTA 1	1	2.0
ASD	35	70.0		Not assessed	37	74.0
ADHD	8	16.0	SP	CSHQ	7	14.0
ASD+ADHD	2	4.0		PSQ	2	4.0
DD	3	6.0	ST	Parent report	1	2.0
LD	1	2.0				
NDDs	1	2.0				

Note. ASD = autism spectrum disorder; ADHD = attention deficit/hyperactivity disorder; DD = developmental disability; LD = learning disability; NDDs = neurodevelopmental disorders; DSM = *Diagnostic and Statistical Manual of Mental Disorders*; GARS = Gilliam Autism Rating Scale; SRS = Social Responsiveness Scale; CBCL = Child Behavior Checklist; WOTA 1 = Water Orientation Test Alyn version 1; CSHQ = Children’s Sleep Habits Questionnaire; PSQ = Pediatric Sleep Questionnaire; PA = physical activity; SP = sleep; ST = screen time. * Based on Piaget’s stages of cognitive development; ** Report measures with significant usage only.

**Table 3 behavsci-15-00592-t003:** Summary of study details included in scoping review *.

Author Location	Neurodiversity **	Study Design	Intervention	Participants	Key Measures	Key Findings
Cai et al. China	ASD	Quasi-experimental	Mini-Basketball Training Program, MBTP (40 min, 5 days per week for 12 weeks)	30 children (26 boys and 4 girls), 15 each in intervention and control groups.	PA: NA ND: CARS SF: SRS	The 12-week MBTP significantly improved social communication abilities in preschool children with ASD.
Cei et al. Italy	ASD	Mixed methods	Football Program (120 min, 2 days per week for 32 weeks)	90 children aged 7–18.	PA: NA ND: DSM SF: ASSM	Quantitative results showed no improvement in social functioning, while qualitative results indicated enhancements in psychosocial skills, including communication and social interaction.
Coffey et al. Ireland	ASD	Quasi-experimental	Integrative exercise (60 min, 3 days per week for 8 weeks)	66 children (55 boys and 11 girls), aged 4–12.	PA: NA ND: GARS SF: ABC	Integrative exercise can significantly reduce social withdrawal and related behaviors in children with severe ASD symptoms.
Dovgan et al. Australia	ASD	Cross-sectional	NA	129 children aged 6–18.	PA: CBCL ND: ADI-R and ADOS SF: VABS and CBCL	In children with ASD, more activity participation was linked to more friendships, while higher IQs correlated with greater internalization of social difficulties.
Firouzjah et al. Iran	ASD	Quasi-experimental	Sensory–motor integration exercises (45 min, 3 days per week for 4 weeks)	30 children (all boys), 15 each in intervention and control groups.	PA: NA ND: doctor’s diagnosis SF: SSRS	Sensory–motor integration exercises significantly improved social skills, including determination, cooperation, and self-control, in children with ASD.
Goldman et al. USA	ASD	Cross-sectional	NA	1784 children, aged 2–18.	SP: CSHQ ND: ADOS SF: PCQ	Sleep problems are strongly linked to daytime behavioral issues, including impaired social functioning, in children with ASD.
Güeita-Rodríguez et al. Spain	ASD	Mixed methods	Water Specific Therapy, WST (60 min, 2 days per week for 28 weeks)	6 children (5 boys and 1 girl), aged 6–12.	PA: NA ND: DSM SF: PSPCSA and PedsQL	Quantitative results showed no improvement in peer or maternal acceptance, while qualitative results noted better social communication and interaction in children with ASD after the WST intervention.
Gundogmus et al. Turkey	ASD	Case-controlled	NA	60 adolescents (34 boys and 26 girls), 30 each in intervention and control groups, aged 10–15.	PA: NA ND: ABC SF: SSAS	Children with ASD who participate in physical activity show better social skills and behaviors compared to those who do not.
Özcan et al. UK	ASD	RCT	Motor Intervention Program, MIP (60 min, 2 days per week for 12 weeks)	34 children (32 boys and 2 girls), 17 each in intervention and control groups, aged 3–6.	PA: NA ND: GARS SF: SSRS	MIP significantly enhanced social skills in children with ASD, particularly in cooperation, self-control, and social interaction.
Haghighi et al. Iran	ASD	RCT	Combined Physical Training, CPT (80 min, 3 days per week for 8 weeks)	16 children (9 boys and 7 girls), 8 each in intervention and control groups, aged 6–10.	PA: NA ND: GARS SF: GARS	CPT significantly enhances social skills in children with ASD by reducing stereotypical behaviors and improving communication.
Hashemi et al. Iran	ASD	Quasi-experimental	Sensory–motor integration training (60 min, 3 days per week for 8 weeks)	50 children (all boys), 25 each in intervention and control groups, aged 7–12.	PA: NA ND: GARS SF: GARS	Sensory–motor integration training significantly improves social interactions and communication skills in children with high-functioning autism.
Heffler et al. USA	ASD	Quasi-experimental	Comprehensive Intervention (60 min, 2 days per week for 24 weeks)	9 children (8 boys and 1 girl), aged 1.5–3.5.	ST: parent report ND: ADOS SF: VABS	Interventions that reduce screen time and increase social interaction can significantly improve the social skills of children with ASD.
Howells et al. Australia	ASD	Quasi-experimental	Auskick (60–90 min, 1 day per week for 6–22 weeks (average 12 weeks))	40 children (37 boys and 3 girls), 19 in intervention and 21 in control group, aged 5–12.	PA: NA ND: DSM SF: VABS-3 and CBCL	Children with ASD in the Auskick program showed significant improvements in social skills, particularly in reducing negative experiences like loneliness and dependency.
Kaur et al. USA	ASD	RCT	Yoga Exercise and Academic Activities (75 min, 4 days per week for 8 weeks)	24 children (21 boys and 3 girls), 12 each in yoga and academic groups, aged 5–13.	PA: NA ND: ADOS SF: SCBE	Creative yoga interventions significantly improve verbal communication, joint attention, and social skills, and reduced negative emotions in children with ASD.
Lee et al. China	ASD	Quasi-experimental	Cooperative physical activities (25 min, 3 days per week for 16–20 weeks)	3 children aged 2–5.	PA: NA ND: DSM SF: teacher report	Cooperative physical activities improve social interactions and may reduce inappropriate behaviors in preschoolers with ASD during physical education and free play.
Lee et al. USA	ASD	Quasi-experimental	Movement-based intervention program (75 min, 2 days per week for 8 weeks)	19 children (16 boys and 3 girls), aged 5–16.	PA: NA ND: doctor’s diagnosis SF: SSIS	Engagement in a movement-based intervention program resulted in significant improvements in physical activity-related social skills among children with ASD.
Marzouki et al. Tunisia	ASD	RCT	Technical Aquatic Training, TAT; Game-Based Aquatic Training, GAT (50 min, 2 days per week for 8 weeks)	22 children (20 boys and 2 girls), 8 each in TAT and GAT groups, and 7 in control group, aged 6–7.	PA: NA ND: DSM SF: GARS	Both types of aquatic training can improve social functioning in children with ASD by reducing stereotypical behaviors and providing opportunities for social interaction.
Memari et al. Iran	ASD	Cross-sectional	NA	68 children (42 boys and 26 girls), aged 6–16.	PA: GT3X monitor Actigraph ND: DSM SF: ASSP	Higher levels of physical activity are associated with better social functioning in children with ASD.
Mohamed et al. Egypt	ASD	RCT	Group exercise (60 min, 3 days per week for 12 weeks)	30 children (24 boys and 6 girls), 15 each in intervention and control groups, aged 8–10.	PA: NA ND: DSM SF: ASSP	Group exercise programs significantly improve social skills in children with ASD.
Morris et al. UK	ASD	Mixed methods	Just Dance: home-based (25 min, 2 days per week for 6 weeks); school-based (10 min, 2 days per week for 6 weeks)	35 children (26 boys and 9 girls), 4 in home-based and 31 in school-based intervention, aged 8–10.	PA: NA ND: DSM SF: ERSSQ	Quantitative results showed improved social communication skills in children with ASD after the Just Dance intervention, while qualitative findings noted its enjoyment and ease of implementation.
Movahedi et al. Iran	ASD	RCT	Kata technique training (30–90 min, 4 days per week for 14 weeks)	30 children (26 boys and 4 girls), 15 each in intervention and control groups, aged 5–16.	PA: NA ND: DSM SF: GARS	Kata technique training improved social skills, emotional regulation, and social engagement in children with ASD.
Najafabadi et al. Iran	ASD	Quasi-experimental	Sports, Play and Active Recreation for Kids, SPARK (40 min, 3 days per week for 12 weeks)	26 children (all boys), 12 in intervention and 14 in control group, aged 5–12.	PA: NA ND: DSM SF: GARS and ATEC	The 12-week SPARK program effectively enhances the social interaction skills of children with autism spectrum disorders.
Narasingharao et al. India	ASD	Quasi-experimental	Structured yoga (75 min, 1 day per week for 12 weeks)	64 children (all boys), 32 each in intervention and control groups, aged 5–16.	PA: NA ND: ICD-10 SF: self-developed questionnaire	A structured yoga intervention improved social skills in children with ASD, reducing aggression and self-injury and enhancing eye contact and social interactions.
Pan et al. China	ASD	Mixed methods	Water Exercise Swimming Program, WESP (45 min, 2 days per week for 10 weeks)	16 children (all boys), 8 each in intervention and control groups, aged 6–9.	PA: NA ND: DSM SF: SSBS	Quantitative findings showed that the WESP improved social interactions and behaviors in children with ASD, while qualitative findings revealed more positive daily social behaviors.
Phung et al. USA	ASD	RCT	Mixed Martial Arts, MMA (45 min, 2 days per week for 13 weeks)	34 children (28 boys and 6 girls), 14 in intervention and 20 in control group, aged 8–11.	PA: NA ND: ADOS and SCQ SF: SSIS	The MMA intervention significantly enhanced social skills and reduced negative social behaviors in boys with ASD.
Qiao et al. China	ASD	Mixed experimental design	Mini-Basketball Training, MBTP (40 min, 5 days per week for 12 weeks)	34 children (29 boys and 5 girls), 19 in intervention group and 15 in control group, aged 3–6.	PA: NA ND: DSM SF: SRS	A 12-week mini-basketball program improves social cognition in preschool children with ASD.
Qi et al. China	ASD	RCT	Ball Combination Training, BCT, and Mini-Basketball Training, MBT (40–45 min, 5 days per week for 12 weeks)	41 children (35 boys and 6 girls), 13 in intervention group 1, 14 in intervention group 2, and 14 in control group, aged 4.23–5.76.	PA: NA ND: doctor’s diagnosis SF: SRS	Children with ASD in BCT and MBT showed greater improvements in social communication than those receiving standard rehabilitation.
Rivera et al. USA	ASD	Mixed methods	Judo Program (45 min, 1 day per week for 8 weeks)	25 children (22 boys and 3 girls), aged 8–17 (12.67).	PA: NA ND: doctor’s diagnosis SF: ABC	Quantitative analysis found no behavioral improvements, while qualitative feedback from parents indicated improvements in social skills and self-esteem.
Sansi et al. Turkey	ASD	Mixed methods	Inclusive Physical Activity, IPA (60 min, 2 days per week for 12 weeks)	45 children (22 boys and 23 girls), 13 in intervention and 9 in control group for ASD, 14 in intervention and 9 in control group for TD, aged 6–11.	PA: NA ND: SSRS SF: SSRS, FAS, and ACL	Quantitative analyses found no improvement in social skills, while qualitative results indicated IPA’s effectiveness in developing social skills in children with ASD.
Toscano et al. Brazil	ASD	NCT	Physical exercise (48 min, 2 days per week for 48 weeks)	229 children (196 boys and 33 girls), 127 in intervention group, 62 in control group 1, and 40 in control group 2, aged 2.3–17.3.	PA: NA ND: DSM SF: ATA	Physical exercise significantly improved social functioning in children and adolescents with ASD, especially in social interaction, attention, and behavior.
Wang et al. China	ASD	Quasi-experimental	Mini-Basketball Training Program, MBTP (40 min, 5 days per week for 12 weeks)	33 children (28 boys and 5 girls), 18 in intervention and 15 in control group, aged 3–6.	PA: NA ND: DSM SF: SRS	MBTP significantly enhanced social communication skills and reduced repetitive behaviors in children with ASD.
Yang et al. China	ASD	Quasi-experimental	Mini-Basketball Training Program, MBTP (40 min, 5 days per week for 12 weeks)	30 children (25 boys and 5 girls), 15 each in intervention and control groups, aged 3–6.	PA: NA ND: DSM SF: SRS	MBTP improved social communication skills in children with ASD, particularly in the social cognition and social communication dimensions.
Yang et al. China	ASD	Quasi-experimental	Mini-Basketball Training Program, MBTP (40 min, 5 days per week for 12 weeks)	30 children (26 boys and 4 girls), 15 each in intervention and control groups, aged 3–6.	PA: NA ND: DSM SF: SRS	MBTP significantly reduced social communication deficits in children with ASD, particularly in social cognition and autistic behavior patterns.
Zanobini et al. Italy	ASD	RCT	Acqua Mediatrice di Comunicazione (30 min, 1 day per week for 12 weeks)	25 children (19 boys and 6 girls), 13 in intervention and 12 in control group, aged 3–8.	PA:NA ND: ABC SF: SRS	The "Acqua Mediatrice di Comunicazione" swimming program enhanced social skills in children with ASD, with benefits sustained six months post-intervention.
Zhao et al. China	ASD	Quasi-experimental	Structured physical activity program (60 min, 2 days per week for 12 weeks)	41 children (29 boys and 12 girls), 21 in intervention and 20 in control group, aged 5–8.	PA: NA ND: DSM SF: SSIS and ABLLS-R	Structured physical activity programs significantly improve social interaction, communication, responsiveness, and expression in children with ASD.
Craig et al. Canada	ADHD	Cross-sectional	NA	192 children, mean age is 10.23.	SP: PSQ ND: SNAP-Iv SF: WFIRS-P	Sleep problems, especially insomnia and daytime sleepiness, significantly affect social functioning in children with ADHD.
Keshavarzi et al. Iran	ADHD	RCT	Sleep training (once a week for 12 weeks)	40 children (38 boys and 2 girls), 20 each in intervention and control groups, aged 8–13.	SP: CSHQ ND: DSM SF: KID-SCREEN-52	A twelve-week sleep training enhances sleep quality and social behavior, as well as emotional and behavioral functioning, in children with ADHD.
Lucas et al. Australia	ADHD	Cross-sectional	NA	257 children (189 boys and 68 girls), aged 5–13.	SP: CSHQ ND: DSM SF: SDQ	In children with ADHD, daytime sleepiness was linked to social difficulties at school, while parent-reported sleep problems showed no significant association.
Majorek et al. Switzerland	ADHD	Quasi-experimental	Therapeutic Eurythmy, TE (30 min, 1 day per week for 36 weeks)	5 children (all boys), aged 8.5–10.	PA: NA ND: doctor’s diagnosis SF: CRS	TE significantly enhances social behavior and reduces hyperactivity symptoms in children with ADHD.
Pan et al. China	ADHD	RCT	Table Tennis Exercise (70 min, 2 days per week for 24 weeks)	32 children (all boys), 16 each in intervention and control groups, aged 6–12 (8.9).	PA: NA ND: DSM SF: CBCL	The table tennis training program significantly reduced social problems and aggression in children with ADHD, with lasting improvements.
Sahin et al. Turkey	ADHD	Cross-sectional	NA	85 children (55 boys and 30 girls), aged 7–12 (9.49).	SP: CSHQ ND: DSM SF: TBAG, FPRT and RMET	Sleep disorders are linked to decreased social cognitive abilities in children with ADHD, potentially affecting social functioning.
Shah et al. India	ADHD	RCT	Sleep training (face-to-face sessions: weeks 2, 4, 8, total of 12 weeks; telephone sessions: 45 min, weeks 1, 3, 6, 10 total, 12 weeks)	100 children, 50 each in intervention and control groups, aged 8–12 (10.66).	SP: CSHQ ND: DSM SF: SDQ and PQLI	The sleep training program significantly improved sleep quality, quality of life, and social, emotional, behavioral, and executive functioning in children with ADHD.
Taylor et al. USA	ADHD	Cross-sectional	NA	3470 children (2376 boys and 1094 girls), aged 6–17 (11.97).	PA: NA ND: NA SF: NA	Meeting the 24 h movement behavior guidelines is linked to improved social cognition and reduced social difficulties in children with ADHD.
Ng et al. USA	ADHD + ASD	Retrospective clinical	NA	114 children (93 boys and 21 girls), aged 7–17 (11.26).	SP: PSQ ND: DSM SF: CBCL	Sleep disorders are strongly linked to behavioral and emotional regulation problems in children with ASD and ADHD.
Zhao et al. USA	ASD + ADHD	Cross-sectional	NA	979 children (795 boys and 184 girls), aged 6–17 (12.3).	PA: NA ND: NA SF: NA	Adhering to 24 h movement guidelines, particularly sleep and screen time, was strongly linked to improved social functioning in children with ASD and ADHD.
Meng et al. Korea	DD	RCT	Service-learning soccer training (90 min, 1 day per week for 12 weeks)	36 children (all boys), 18 each in intervention and control groups, mean age 11.5.	PA: NA ND: doctor’s diagnosis SF: KBASC	The service-learning soccer training program helps improve the social skills and self-regulation of children with developmental disabilities.
Oh et al. Korea	DD	Mixed methods	Parent-led Aquatic Activities (60 min, 2 days per week for 8 weeks)	14 children (9 boys and 5 girls), aged 9–15 (11.07).	PA: NA ND: doctor’s diagnosis SF: KNISE-SAB	Quantitative results showed no impact of parent-led water activities on social behaviors, while qualitative findings highlighted parental involvement in boosting resilience.
Schoen et al. USA	DD	Mixed methods	Bicycle Riding Training (60 min per week for 5 days)	19 children (12 boys and 7 girls), aged 4.43–10.6 (6.49).	PA: NA ND: SP3D SF: SP3D and VAS	The Bicycle Riding Program enhances the social skills and self-esteem of children with developmental disabilities.
Z. Liu et al. USA	LD	Cross-sectional	NA	4999 children aged 6–17 (12.0, gender not reported).	PA: NA ND: NA SF: NA	Adherence to 24 h movement guidelines is linked to improved social functioning and emotional wellbeing in children with learning disabilities.
Kara et al. Turkey	NDDs	Prospective case-control	NA	317 children (211 boys and 106 girls), 166 in case and 151 in control group, aged 4–6.	SP: CSHQ ND: DSM SF: SCBE	Sleep disturbances in children with neurodevelopmental disorders are strongly linked to social deficits and behavioral issues, including aggression and anxiety.

Note: * Detailed references of the eligible studies can be found in the Appendix A. ‘List of included studies’ in the Appendix A. ** all neurodiverse children have been diagnosed; RCT = randomized controlled trial; NCT = non-randomized controlled trial; ASD = autism spectrum disorder; ADHD = attention deficit/hyperactivity disorder; DD = developmental disability; LD = learning disability; NDDs = neurodevelopmental disorders; PA = physical activity; SP = sleep; ST= screen time; ND = neurodiversity; SF = social function; CARS = Childhood Autism Rating Scale; SRS = Social Responsiveness Scale; DSM = *Diagnostic and Statistical Manual of Mental Disorders;* ASSM = Adaptive Social Skills Measure; GARS = Gilliam Autism Rating Scale; ABC = Aberrant Behavior Checklist; VABS = Vineland Adaptive Behavior Scale; CBCL = Child Behavior Checklist; ADI-R = Autism Diagnostic Interview—Revised; ADOS = Autism Diagnostic Observation Schedule; SSRS = Social Skills Rating System; CSHQ = Children’s Sleep Habits Questionnaire; PCQ = Parental Concerns Questionnaire; PSPCSA = Pictorial Scale of Perceived Competence and Social Acceptance for Young Children; PedsQL = Pediatric Quality of Life Inventory; ABC = Autism Behavior Checklist; SSAS = Social Skills Assessment Scale; SSRS = Social Skills Rating System; SSIS = Social Skills Improvement System; ASSP = Autism Social Skills Profile; ERSSQ = Emotional Regulation and Social Skills Questionnaire; ATEC = Autism Treatment Evaluation Checklist; SSBS = School Social Behavior Scale; FAS = Friendship Activity Scale; ACL = Adjective Checklist; ATA = Autistic Traits Assessment Scale; ABLLS-R = Assessment of Basic Language and Learning Skills—Revised; PSQ = Pediatric Sleep Questionnaire; SNAP-Iv = Swanson, Nolan and Pelham-IV; WFIRS = Weiss Functional Impairment Rating Scale; KID-SCREEN 52 = KIDSCREEN-52 Quality of Life Questionnaire; SDQ = Strengths and Difficulties Questionnaire; CRS = Conner’s Rating Scale; FPRT = Faux Pas Recognition Test; RMET = Reading the Mind in the Eyes Test; PQLI = Pediatric Quality of Life Inventory; SCBE = Social Competence Behavior Evaluation; KBASC = Korean Behavior Assessment System for Children; KNISE-SAB = Korea National Institute of Special Education—Scales of Adaptive Behavior; SP3D = Sensory Processing 3 Dimensions; VAS = Visual Analog Scale; SCBE = Social Competence and Behavior Evaluation; NA = not assessed.

## Data Availability

No new primary data were created or analyzed in this study.

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
