# Peer review of "Twenty-Four-Hour Movement Behaviors and Social Functions in Neurodiverse Children: A Scoping Review"

_behavsci, 2025, doi:10.3390/bs15050592_

Round 1

Reviewer 1 Report

Comments and Suggestions for Authors

The introduction is grounded in current research evidence, including recent studies, and effectively frames the rationale for the review. The Methods section is clearly written, well-structured, and follows established scoping review frameworks. No major issues were identified, and the section is satisfactory as written. The Results section provides a comprehensive and well-organized synthesis of the included studies, with clear subgroup analyses based on types of neurodiversity, movement behaviors, and intervention designs. The tables and figures presented in the paper are appropriate, clearly formatted, and contribute to the reader’s understanding of the findings. Their inclusion enhances the clarity and accessibility of the data. The Discussion section offers a rich interpretation of the review’s findings and appears to be the strongest part of the paper. The authors effectively contextualize their results within existing literature, emphasizing the importance of incorporating all components of 24-hour movement behavior into research and practice. The discussion of intervention design is both practical and well-supported. Overall, this section is well-constructed and offers direction for future research and interventions.

Minor refinements could further strengthen clarity and impact, but no major revisions are necessary.

More specific:

  • Line 49: The phrase “for alleviating maladaptiveness in neurodiverse children” is unclear and may be difficult for readers to interpret. Please consider rephrasing or clarifying what is meant by “maladaptiveness” in this context.
  • In the introduction section, the authors should strengthen the transition from general mental health benefits to the specific focus on social function.
  • While the inclusion of qualitative evidence is appropriate and well-justified, the discussion would benefit from briefly referencing a specific example where qualitative data revealed improvements in social functioning that were not evident in the quantitative results. This would strengthen the authors’ argument for the value of mixed-methods approaches.

The paper presents a well executed and timely scoping review on the relationship between 24-hour movement behaviors, physical activity, sedentary behavior, and sleep, and social functioning in neurodiverse children. The topic is comprehensive and timely, filling a gap in current knowledge. The review moves beyond physical activity to also consider sedentary behavior and sleep, which is innovative and reflects recent developments in behavioral health research. The literature cited is recent, relevant, and appropriately diverse, with no evidence of excessive self-citation. The authors clearly explain the limitations of existing research, including the lack of studies addressing multiple movement behaviors, the underuse of qualitative and mixed methods, and the narrow focus on ASD at the expense of other neurodiverse populations. The review is methodologically sound, following established scoping review guidelines and reporting practices. It contributes meaningful insights to the field of child development, particularly by advocating for more inclusive research approaches. Overall, this is a valuable and well structured paper that needs only minor revisions to enhance clarity and strengthen a few interpretive points.

Author Response

Dear Editor and Reviewers,

Thank you for reviewing and considering the manuscript titled "24-hour Movement Behaviors and Social Functions in Neurodiverse Children: A Scoping Review." We appreciate your recognition, positive comments and constructive suggestions. In this revision, we have carefully addressed each of your points which further improved the quality of the manuscript. Please find below our responses to each of your comments alongside with the summary of revisions made. We hope that the revision has met your standards and look forward to the acceptance of publication of the work.

Reviewer 1:

General comment: The introduction is grounded in current research evidence, including recent studies, and effectively frames the rationale for the review. The Methods section is clearly written, well-structured, and follows established scoping review frameworks. No major issues were identified, and the section is satisfactory as written. The Results section provides a comprehensive and well-organized synthesis of the included studies, with clear subgroup analyses based on types of neurodiversity, movement behaviors, and intervention designs. The tables and figures presented in the paper are appropriate, clearly formatted, and contribute to the reader’s understanding of the findings. Their inclusion enhances the clarity and accessibility of the data. The Discussion section offers a rich interpretation of the review’s findings and appears to be the strongest part of the paper. The authors effectively contextualize their results within existing literature, emphasizing the importance of incorporating all components of 24-hour movement behavior into research and practice. The discussion of intervention design is both practical and well-supported. Overall, this section is well-constructed and offers direction for future research and interventions.

Response: Thank you for your time reviewing the manuscript and providing positive, constructive comments. In this revision, we have fully addressed each of your comments, and have made changes accordingly. We provide our responses to each point below. Your comments and recommendations have helped us strengthen the manuscript in a meaningful way, and thus we very much appreciate your contribution. Thank you.

Comment 1.1: Line 49: The phrase “for alleviating maladaptiveness in neurodiverse children” is unclear and may be difficult for readers to interpret. Please consider rephrasing or clarifying what is meant by “maladaptiveness” in this context.

Response 1.1: Thank you for your insightful comment. In response, we have clarified the phrase "maladaptiveness" in the manuscript. Specifically, we have revised it to better describe the difficulties related to social functioning and behavioral challenges in neurodiverse children. We believe this revision enhances the clarity of the text; see Lines 48-50).

Comment 1.2: In the introduction section, the authors should strengthen the transition from general mental health benefits to the specific focus on social function.

Response 1.2: Thank you for the suggestion. We fully agreed it. In this revision, we justified the relevance between social functioning and mental health and strengthened the flow linking social function to one’s general mental health; see Lines 52-54.

Comment 1.3: While the inclusion of qualitative evidence is appropriate and well-justified, the discussion would benefit from briefly referencing a specific example where qualitative data revealed improvements in social functioning that were not evident in the quantitative results. This would strengthen the authors’ argument for the value of mixed-methods approaches.

Response 1.3: Thank you for your comment. We fully agree with your point that adding a specific example in the discussion would strengthen our argument for the value of mixed-methods research. In this revision, we have included a concrete example of a mixed-methods study and highlighted the advantages and value of qualitative methods in complementing quantitative data. Please refer to lines 511-519 for the details. We hope our revision meets your expectations. Of course, if you feel further elaboration is needed, we would be happy to make additional adjustments.

Reviewer 2 Report

Comments and Suggestions for Authors

The review addresses an emerging issue – the impact of 24-hour movement behaviors (physical activity, sedentary behavior, sleep) on social function in neurodivergent children. It is an important contribution to synthesis: it highlights the gaps in the current literature (little attention to the combination of the three behaviors, poor use of qualitative approaches, and lack of longitudinal follow-ups).

This is an excellent work of synthesis on a very relevant topic with concrete implications for intervention in neurodivergent children. With some revisions aimed at the clarity of exposition, interpretative synthesis, and exploitation of practical implications, this work has the credentials of a quality publication.

1. The language is generally clear, but there are redundant or overly complex sentences, especially in the introduction and discussion.

Example: “Such a conceptualization embraced the approach of not only focusing on overcoming the weaknesses but also enhancing the strengths…” you could simplify and make the wording more direct.

We suggest reducing verbosity and aiming for more concise sentences while maintaining scientific precision.

2. The contribution is original, as it extends the concept of “physical activity” to the entire spectrum of 24-hour movement behaviors, in the context of childhood neurodivergence.

 However, a clear central research question is missing, explicitly stated at the beginning.

At the end of the introduction, insert an explicit guiding question such as:

“This review aims to answer: how do 24-hour movement behaviors affect social functioning in neurodiverse children?”

  1. Methodology

The methodology is adequately described and adheres to international standards. The inclusion of the PRISMA-ScR criteria is a very positive point.

However, the quality of the included studies is not assessed, which may make it difficult to interpret the robustness of the conclusions.

The ways of classifying neurodiversity could be better defined (ASD, ADHD, LD, etc.).

  1. Results

Very detailed, and supported by comprehensive tables. The summary is however very dense and in some places difficult to follow.

Consider adding graphic summaries (e.g. summary diagram by type of intervention and impact on social function) and strengthen the paragraphs that interpret the data rather than just describe them.

  1. Discussion and Practical Implications

The discussion is solid but at times repetitive. In addition, there is no specific reference to the educational, clinical, or sports context, where this evidence could be applied.

Include a subsection “Implications for practice” at the end of the discussion, indicating how professionals (teachers, therapists, coaches) can apply these results.

  1. Limitations

Some limitations are acknowledged (little methodological variety, lack of longitudinal studies, etc.). However, there is no mention of publication bias and the possible overrepresentation of positive results. Insert a sentence like:

“The potential for publication bias should be acknowledged, as studies reporting null or negative findings may be underrepresented.”

Author Response

Dear Editor and Reviewers,

Thank you for reviewing and considering the manuscript titled "24-hour Movement Behaviors and Social Functions in Neurodiverse Children: A Scoping Review." We appreciate your recognition, positive comments and constructive suggestions. In this revision, we have carefully addressed each of your points which further improved the quality of the manuscript. Please find below our responses to each of your comments alongside with the summary of revisions made. We hope that the revision has met your standards and look forward to the acceptance of publication of the work.

Reviewer 2:

General comment: The review addresses an emerging issue – the impact of 24-hour movement behaviors (physical activity, sedentary behavior, sleep) on social function in neurodivergent children. It is an important contribution to synthesis: it highlights the gaps in the current literature (little attention to the combination of the three behaviors, poor use of qualitative approaches, and lack of longitudinal follow-ups).

This is an excellent work of synthesis on a very relevant topic with concrete implications for intervention in neurodivergent children. With some revisions aimed at the clarity of exposition, interpretative synthesis, and exploitation of practical implications, this work has the credentials of a quality publication.

Response: Thank you for your positive feedback. We appreciate your recognition of the importance of our review and the gaps it highlights in the literature. Your suggestions for improving clarity, synthesis, and practical implications are valuable, and we will incorporate them to further strengthen our work. Thank you again for your thoughtful comments.

Comment 2.1: The language is generally clear, but there are redundant or overly complex sentences, especially in the introduction and discussion.

Example: “Such a conceptualization embraced the approach of not only focusing on overcoming the weaknesses but also enhancing the strengths…” you could simplify and make the wording more direct.

We suggest reducing verbosity and aiming for more concise sentences while maintaining scientific precision.

Response 2.1: Thank you for the comment. We agreed that condensing overly complex sentences can enhance readability of the texts. We have followed your suggestions accordingly when making revisions (e.g., see Lines 62-66, 90-92). We have also proofread the entire manuscript thoroughly and made changes for conciseness. Should you have more specific suggestions for improvement, we are happy to make further changes.

Comment 2.2: The contribution is original, as it extends the concept of “physical activity” to the entire spectrum of 24-hour movement behaviors, in the context of childhood neurodivergence.

However, a clear central research question is missing, explicitly stated at the beginning.

At the end of the introduction, insert an explicit guiding question such as:

“This review aims to answer: how do 24-hour movement behaviors affect social functioning in neurodiverse children?”

Response 2.2: Adding this sentence at the end of the introduction does help clarify and highlight the main research question or the primary aim. Following your suggestion, we have made changes accordingly. See lines 112-114.

Comment 2.3:

Methodology:

The methodology is adequately described and adheres to international standards. The inclusion of the PRISMA-ScR criteria is a very positive point.

However, the quality of the included studies is not assessed, which may make it difficult to interpret the robustness of the conclusions.

The ways of classifying neurodiversity could be better defined (ASD, ADHD, LD, etc.).

Response 2.3: Thank you for your comments on the methodology section of our manuscript. We appreciate your recognition of our description of the methodology and the use of the PRISMA-ScR tool. We acknowledge that assessing the quality of the included studies would enhance the rigor of the research. However, we followed Arksey and O'Malley’s five key stages of the scoping review process, which does not suggest conducting a quality assessment. According to Arksey and O'Malley’s guidance and the PRISMA-ScR, the primary aim of a scoping review is to explore the breadth of existing research, its trends, and any knowledge gap to inform theory, practices, and future directions. In the current scoping review, we did not generate conclusions on regarding the magnitude of relationship between 24-hour movement behaviour and social functioning in neurodiverse children. As such, our findings and conclusions should not be biased due to not conducting a quality assessment of the included studies. However, we acknowledge the benefit of conducting a quality assessment, as this would better address potential heterogeneity in the studies. In this revision, we have therefore strengthend the discussion of this point in the limitation sections; see lines 546-554.

Additionally, we would like to clarify that the classification of neurodiverse children depends on the relevant measurement tools used in the included studies. To avoid risk confusing the classifications of ADHD, ASD and other neurodiversity traits in the already rich literature, we decided to specify the measures used in the included studies (see Table 3 note) for interested readers.

Comment 2.4:

Results:

Very detailed, and supported by comprehensive tables. The summary is however very dense and in some places difficult to follow.

Consider adding graphic summaries (e.g. summary diagram by type of intervention and impact on social function) and strengthen the paragraphs that interpret the data rather than just describe them.

Response 2.4: Thank you very much for recognizing our efforts in results synthesis and reporting, including the comprehensive presentation of the table contents. Since we followed the principle of broad synthesis which is key to scoping review, we did not commit to comparing effects of different types of interventions. This is also because different social function measures were used in varied interventions and the number of different types of interventions included in this review varied. Providing a general summary rather than specific comparisons of findings from the included studies is recommended practices for scoping review. We also recognize that we did not perform quality assessment for included studies, nor attempted to examine and make conclusion about the magnitude of any effect (otherwise a systematic review or meta-analysis would fit). According to Arksey and O'Malley’s guidance and the PRISMA-ScR, results synthesis of a scoping review should be descriptive in its nature. We believe that we have followed the guidelines closely and prepared the summary of study details table in accordance with best practices. However, the reviewer can provide more specific recommendations for change, or offer any examples change suggestions, we are more than happy to make further revisions.

Comment 2.5:

Discussion and Practical Implications:

The discussion is solid but at times repetitive. In addition, there is no specific reference to the educational, clinical, or sports context, where this evidence could be applied.

Include a subsection “Implications for practice” at the end of the discussion, indicating how professionals (teachers, therapists, coaches) can apply these results.

Response 2.5: We thank and respect this comment from Reviewer 2 and fully agreed the importance of highlighting practical implications to readers based on the scoping review findings. However, based on our experience and also the observation of common practice in articles published in the Behavioral Sciences (and many other scientific publications), implications for practice rarely stands alone at the end of the discussion section. In fact, PRISMA guidelines and APA’s journal reporting standards (JARS) encourage authors to integrate practical implications to the discussion of key findings. We, therefore, have followed this recommended approach and further strengthened implications for practices in our manuscript (i.e., the suggestion of the reviewer).

Specifically, we emphasized in the opening of the discussion section that the section would be focusing on communicating key takeways and implications for practices from the scoping review findings (see lines 404-405). We then integrated our suggestions for practitioners and future practices based on the four key scoping review findings discussed in section 4.1 (see practical implications in lines 432-434), section 4.2 (see practical implications in lines 450-452, 466-468), section 4.3 (see practical implications in lines 490-492), and section 4.4 (see practical implications in lines 550-555). More importantly, we have strengthened the conclusion section by providing more direct takeaway messages for future applications (see lines 517-522). We believe these re-emphasized practical implications added to this revision align with the reviewer’s suggestions.

Comment 2.6:

Limitations

Some limitations are acknowledged (little methodological variety, lack of longitudinal studies, etc.). However, there is no mention of publication bias and the possible overrepresentation of positive results. Insert a sentence like:

“The potential for publication bias should be acknowledged, as studies reporting null or negative findings may be underrepresented.”

Response 2.6: Thank you for the suggestion. We have made the requested changes accordingly; see lines 551-554.

Reviewer 3 Report

Comments and Suggestions for Authors

In this research study, authors conducted a systematic search via four databases using Web of Science, PubMed, Scopus, and EBSCOhost through 31st December 2024 using scoping review guidelines for this synthesis. Results indicated that Initial search provided 2,342 articles including 50 studies fulfilled the inclusion criteria after a strong screening process. They found that research studies were published between 2004 and 2024, basically focused on children with neurodiversity of Autism Spectrum Disorder or ASD (70%), Attention Deficit Hyperactivity Disorder or ADHD 20 (23%), and quantitative methods (84%).

On the other hand, only 6% of studies studied the combined effects of different 24-hour movement behaviors, with majority examining physical activity alone. In addition, longer group exercise interventions were found to be more effective in improving social functions in neurodiverse children. In summary,  Authors found that 24-hour movement behaviors had a positive influence on the social functioning of neurodiverse children.

I would like to thank authors for this interesting research study. Exercise and physical activity seem an effective way of intervention for children with autism. I believe authors made such an organized scoping review and they had reliability and validity as well.

However, authors need to write their hypothesis in a clear way. In addition, this study had a weak conclusion section and this section needs improvement with future practical applications. I look forward to seeing edited version this manuscript. Thank you.

Author Response

Dear Editor and Reviewers,

Thank you for reviewing and considering the manuscript titled "24-hour Movement Behaviors and Social Functions in Neurodiverse Children: A Scoping Review." We appreciate your recognition, positive comments and constructive suggestions. In this revision, we have carefully addressed each of your points which further improved the quality of the manuscript. Please find below our responses to each of your comments alongside with the summary of revisions made. We hope that the revision has met your standards and look forward to the acceptance of publication of the work.

Reviewer 3:

Comment: In this research study, authors conducted a systematic search via four databases using Web of Science, PubMed, Scopus, and EBSCOhost through 31st December 2024 using scoping review guidelines for this synthesis. Results indicated that Initial search provided 2,342 articles including 50 studies fulfilled the inclusion criteria after a strong screening process. They found that research studies were published between 2004 and 2024, basically focused on children with neurodiversity of Autism Spectrum Disorder or ASD (70%), Attention Deficit Hyperactivity Disorder or ADHD 20 (23%), and quantitative methods (84%).

On the other hand, only 6% of studies studied the combined effects of different 24-hour movement behaviors, with majority examining physical activity alone. In addition, longer group exercise interventions were found to be more effective in improving social functions in neurodiverse children. In summary, Authors found that 24-hour movement behaviors had a positive influence on the social functioning of neurodiverse children.

I would like to thank authors for this interesting research study. Exercise and physical activity seem an effective way of intervention for children with autism. I believe authors made such an organized scoping review and they had reliability and validity as well.

However, authors need to write their hypothesis in a clear way. In addition, this study had a weak conclusion section and this section needs improvement with future practical applications. I look forward to seeing edited version this manuscript. Thank you.

Response: Thank you for your positive feedback and valuable suggestions. We appreciate your positive view of this scoping review. In response to your comment about the hypothesis, we do not believe a specific hypothesis fits scoping review based on the nature of the review as well as Arksey and O'Malley’s guidance or the PRISMA-ScR. However, we received the reviewer’s viewpoint of specifying the aim of the review and thus elaborate on our research objectives at the end of the Introduction (also suggested by Reviewer 2; see lines 112-114).

Besides, we have strengthened our discussion section by incorporating more direct and emphasized implications for practices to each of the key takeaway messages thus a stronger synthesis of key findings and concluding remarks for future practices (sections 4.1-4.4; see lines 432-434, 450-452, 466-468, 490-492, 517-522). More importantly, we have strengthened the conclusion section by providing more direct takeaway messages for future applications (see lines 561-569). We hope that you will be happy with our revision and consider this work for acceptance of publication in the journal.